# Nitric Oxide Negatively Regulates the Rapid Formation of *Pleurotus ostreatus* Primordia by Inhibiting the Mitochondrial *aco* Gene

**DOI:** 10.3390/jof8101055

**Published:** 2022-10-08

**Authors:** Ludan Hou, Chenyang Huang, Xiangli Wu, Jinxia Zhang, Mengran Zhao

**Affiliations:** 1Institute of Agricultural Resources and Regional Planning, Chinese Academy of Agricultural Sciences, Beijing 100081, China; 2Key Laboratory of Microbial Resources, Ministry of Agriculture and Rural Affairs, Beijing 100081, China; 3College of Food Science and Engineering, Shanxi Agricultural University, Jinzhong 030801, China; 4Shanxi Key Laboratory of Edible Fungi for Loess Plateau, Jinzhong 030801, China

**Keywords:** *Pleurotus ostreatus*, nitric oxide, aconitase, hydrogen peroxide, fruiting body development

## Abstract

Nitric oxide (NO) is as a signaling molecule that participates in the regulation of plant development and in a number of physiological processes. However, the function and regulatory pathway of NO in the growth and development of edible mushrooms are still unknown. This study found that NO played a negative role in the transformation of *Pleurotus ostreatus* from vegetative growth to reproductive growth by the exogenous addition of NO donors and scavengers. Further studies showed that NO can inhibit the gene expression and enzyme activity of aconitase (ACO). Moreover, the overexpression (OE) of mitochondrial *aco* and RNA interference (RNAi) confirmed that ACO participates in the regulation of the primordia formation rate. The effects of *aco* OE and RNAi on the tricarboxylic acid (TCA) cycle and energy metabolism were further measured. The results showed that RNAi-*aco* mutant strains can affect the enzyme activities of isocitrate dehydrogenase of mitochondria (ICDHm) and α-ketoglutarate dehydrogenase (α-KGDH) in the TCA cycle, thereby reducing the production of nicotinamide adenine dinucleotide (NADH) in the TCA cycle, decreasing the contents of adenosine triphosphate (ATP) and hydrogen peroxide (H_2_O_2_), and negatively regulating the rapid formation of primordia. In addition, H_2_O_2_ was significantly increased during the transformation from vegetative growth to reproductive growth of *P. ostreatus*. Additionally, the exogenous addition of H_2_O_2_ and its scavengers further confirmed the positive regulation by H_2_O_2_ in primordia formation. This study shows that during the growth and development of *P. ostreatus*, NO can inhibit the expression of the mitochondrial *aco* gene and ACO protein in the TCA cycle, reduce the production of ATP and H_2_O_2_ in the respiratory chain, and negatively regulate the rate of primordia formation.

## 1. Introduction

*Pleurotus ostreatus* is a typical heterothallic edible fungus that is widely cultivated all over the world [1,2]. It is rich in nutrients and has economic and ecological value and medicinal properties [3]. Recently, the growth and development of mushrooms has become a trending topic in mycological research [4]. Many functional genes, transcription factors, and signal transduction pathways related to mushroom development have been found and studied in edible fungi. In *Ganoderma lucidum*, nicotinamide adenine dinucleotide phosphate oxidase genes (*NoxA* and *NoxB*) can not only regulate mycelial branching, fruiting development, and the production of reactive oxygen species (ROS), but also participate in the regulation of ganoderic acid biosynthesis [5]. In *Coprinus cinereus*, the dark stipe1 (*dst1*) gene encodes a putative photoreceptor for blue light, which is involved in the photomorphogenesis of the mushroom [6]. In *Volvariella volvacea*, the glutathione S-transferase encoding gene (*vv-gto1*) can form different splices to affect protein function and then play a role in the special biological function of heterokaryotic hyphae [7]. In addition, cytochrome P450 genes [8], superoxide dismutase (SOD) [9], phenylalanine ammonia-lyase, zinc finger transcription factor, and the cyclic adenylate signaling pathway [10] have also been confirmed to play a crucial role in mushroom development.

At present, many mushroom genomes have been published, including the *P. ostreatus* genome. In addition, some researchers have determined the transcriptome of *P. ostreatus* at different developmental stages, laying a foundation for further research in the future [11].

Nitric oxide (NO) is not only a bioactive free radical, but also a signaling molecule involved in the regulation of a variety of physiological functions. Studies have shown that NO can play a role in vegetative growth in processes of plant growth and development, such as promoting seed germination [12], root growth, leaf expansion, and stomatal closure [13]. It can also affect the degree of fruiting in the reproductive growth of plants. Finally, NO can also participate in the metabolic processes of plant cells, including mitochondrial activity and iron metabolism [14,15]. However, whether NO plays a role in edible fungi and whether it can participate in the development of edible fungi have not been further studied. At present, some studies have shown that NO plays an important regulatory role in fungi [16]. For example, in *Aspergillus*, NO is involved in the formation of conidia [17]. In *Stemphylium eturmiunum*, NO is essential in the formation of conidia and pseudosheaths [18]. In *P. ostreatus*, previous results showed that NO enhanced the response of mycelia to heat stress by regulating the formation of ROS. However, the function and regulatory pathway of NO in the growth and development of *P. ostreatus* are still unknown.

Previous studies have shown that NO can participate in the abiotic stress response of *P. ostreatus* by regulating aconitase (ACO) at the protein and gene levels [19]. ACO (EC 4.2.1.3) not only catalyzes the mutual transformation of citrate and isocitrate in the tricarboxylic acid (TCA) cycle, but also monitors and senses iron homeostasis. There are two subtypes of ACO. The mitochondrial subtype is an integral part of the citric acid cycle. Another subtype exists in the cytoplasm and participates in the glyoxylate cycle [20]. All ACO proteins have unstable [4Fe-4S] clusters [21]. ACO is a type of bifunctional protein. When iron is sufficient, [4Fe-4S] clusters are relatively stable, and ACO has enzymatic activity. However, because [4Fe-4S] clusters are unstable, when organisms are under oxidative stress, [4Fe-4S] clusters are decomposed and lose enzymatic function, but they can perform another function. For example, in *E. coli*, ACO not only has enzymatic activity, but also acts as a posttranscriptional regulator by binding to specific sites on its mRNA [22]. ACO has multiple functions in organisms [23]. In *Arabidopsis* and *Nicotiana benthamiana*, ACO can participate in cell oxidative stress, and plays an important role in mediating oxidative stress and regulating cell death [24]. In addition, ACO is involved in plant growth and development. For example, the reduced expression level of the *aco* gene affects the photosynthetic rate of tomato leaves [25]. In *Salmonella enterica*, the movement ability of flagella decreases in the *aco* mutant strain [26]. In *Streptomyces viridochromogenes*, aerial mycelia and spores could not be formed in the *aco* gene mutant [27]. However, the function of ACO in the growth and development of edible fungi is still unclear.

In this study, the effect of NO on the development of *P. ostreatus* primordia was studied by adding NO donors or scavengers. On this basis, the inhibitory effect of NO on ACO in the TCA cycle was further confirmed. Subsequently, ACO mutants were used to identify the effect of ACO on the formation of primordia. The effects of ACO on the TCA cycle and energy production were explored, and the regulatory effect of ACO mutants on hydrogen peroxide (H_2_O_2_) was verified. Finally, the exogenous addition of H_2_O_2_ and H_2_O_2_ scavengers further confirmed the role of H_2_O_2_ in the formation of *P. ostreatus* primordia. This study preliminarily explored the regulatory pathway by which NO regulates the formation of *P. ostreatus* primordia.

## 2. Materials and Methods

### 2.1. Strains

The *P. ostreatus* CCMSSC 00389 strain was provided by the Center for Mushroom Spawn Standards and Control of China. In DDBJ/EMBL/GenBank, the genome of the CCMSSC 00389 strain is available under the accession number, MAYC00000000 [28]. Overexpression (OE) and RNA interference (RNAi) plasmids were preserved in the laboratory. The OE-*aco* strains and RNAi strains were mutants constructed in previous studies [19].

### 2.2. Mushroom Production

The wild-type (WT), OE-*aco*, and RNAi-*aco* strains were incubated on potato dextrose agar (PDA) medium 3 times. Subsequently, the culture materials were prepared according to previous reports [29]. The cultivation materials were placed into the cultivation bottle so that the weight of each cultivation bottle was 200 g. The culture bottle was sterilized at 126 °C for 120 min. After the culture bottles were cooled to room temperature, the activated strain was inoculated into the culture material. The inoculated culture bottles were cultured at 25 °C in the dark so the mycelium could grow fully. Then, they were transferred to the intelligent mushroom box for mushroom production management.

### 2.3. Determination of NO Content

Intracellular NO content was measured using the NO assay kit (Beyotime, Shanghai, China). First, the mycelium was frozen in liquid nitrogen and ground. Then, 20 mg was placed into a centrifuge tube, and 200 µL of lysate was added. After incubation in an ice bath for 10 min and centrifugation at 14,000× *g* for 10 min, the supernatant was collected. The NO concentration was detected according to the instructions of the kit.

### 2.4. Experiment with the Addition of Exogenous Sodium Nitroprusside (SNP) or 2-(4-Carboxyphenyl)-4,4,5,5-Tetramethylimidazoline-1-Oxyl-3-Oxide (cPTIO)

In this experiment, the culture bottles were divided into 4 groups, with 10 bottles per group. Fruiting body was induced in the end of spawn running. One group was used as the control, and 1 mL of SNP (100 μM) or cPTIO (250 μM) was added to the two other groups. The same amount of SNP + cPTIO was added to the last group. After treatment for 24 h, the exogenous additive was removed. The formation rate of primordia in different treatment groups was observed and photographed.

### 2.5. Determination of ACO Activity

The protein concentration in different samples was determined by a Bradford protein quantification kit (Vazyme, Nanjing, China). The ACO activity in the cytoplasm and mitochondria in different samples was determined using an ACO activity detection kit (Solarbio, Beijing, China).

### 2.6. H_2_O_2_ Content Determination

The protein concentration and H_2_O_2_ content in different samples were determined by a Bradford protein quantification kit (Vazyme, Nanjing, China) and H_2_O_2_ quantitative assay kit (Sangon Biotech, Shanghai, China).

### 2.7. Measurement of the Activities of Isocitrate Dehydrogenase of Mitochondria (ICDHm) and α-Ketoglutarate Dehydrogenase (α-KGDH)

ICDHm catalyzes the formation of α-ketoglutarate from isocitric acidin in the TCA cycle, and this process simultaneously reduces NAD^+^ to NADH. ICDHm is one of the rate-limiting enzymes of the TCA cycle, and its catalytic reaction is one of the main sources of cellular NADH. The α-KGDH is widely distributed in the mitochondria of animals, plant microorganisms, and cultured cells. It is one of the key enzymes regulating the α-oxidative decarboxylation of ketoglutarate to succinyl coenzyme in the TCA cycle [30,31]. The activity of ICDHm and α-KGDH in each sample was detected according to the instructions of the kit (Solarbio, Beijing, China).

### 2.8. NADH and NAD^+^ Content Determination

The NADH content and NAD^+^/NADH ratio in different samples were determined with an NAD^+^/NADH assay kit (Beyotime, Shanghai, China).

### 2.9. Adenosine Triphosphate (ATP) Content Determination

The enhanced ATP assay kit (Beyotime, Shanghai, China) was used to measure the changes in ATP content in differently treated samples.

### 2.10. Western Blot Analysis

According to a previous study [32], the expression of mitochondrial ACO protein in *P. ostreatus* was detected by Western blot. ACO and glyceraldehyde 3-phosphate dehydrogenase (GAPDH) antibodies were synthesized by a company (GenScript, Nanjing, China). Briefly, total protein was extracted from different samples, and the protein concentration was determined by a detergent-compatible Bradford protein quantification kit (Vazyme, Nanjing, China). Then, 20 μg of total protein from different samples was separated on a 12% (*w*/*v*) sodium dodecyl sulfate–polyacrylamide gel electrophoresis gel. After electrophoresis, the proteins were transferred to polyvinylidene fluoride membranes. Finally, ACO antibodies were used for Western blot analysis, with GAPDH antibodies as a reference. The chemiluminescent signal was revealed using ECL substrate for detection [33].

### 2.11. Quantitative Real-Time PCR (qPCR)

RNA was extracted as previously described [34]. The expression of *aco* in differently treated samples was detected according to Wang et al. [29]. In brief, total RNA was first extracted from different samples, and cDNA was synthesized for qPCR analysis. In this study, the *β-actin* gene was used as a reference. The qPCR amplification procedure was as follows: 95 °C for 3 min, 40 cycles of 95 °C for 3 s, and 60 °C for 32 s, and a final extension at 72 °C for 30 s. The relative expression level of genes was calculated according to the 2^−^^ΔΔCT^ method. The primers are shown in Table 1.

### 2.12. Experiment with the Addition of Exogenous H_2_O_2_ or N,N′-Dimethylthiourea (DMTU)

The culture bottles were divided into 5 groups, with 10 bottles per group. Fruiting body was induced when the spawn running completed. One group served as the control (adding 1 mL of H_2_O), and 1 mL of 25 mM or 50 mM H_2_O_2_ was added to the two other groups. Then, 1 mL of 25 mM or 50 mM DMTU was added to the final two groups. After treatment for 24 h, the exogenous additive was removed. The formation rate of primordia in different treatment groups was observed and photographed.

### 2.13. Data Analysis

GraphPad Prism 6 (GraphPad Software, Inc., San Diego, CA, USA) and SPSS statistics 17.0 (SPSS Inc., Chicago, IL, USA) were used for statistical analysis. The values were reported as the means ± SEs, and were analyzed by one-way ANOVA according to Duncan’s test. A *p* value < 0.05 was considered significant.

## 3. Results

### 3.1. NO Content in P. ostreatus Varies in Different Developmental Stages

NO, as a signaling molecule, plays an important role in the growth and development of many organisms. In this study, the changes in NO content in different developmental stages were explored. Figure 1A shows the different developmental stages (mycelia, primordia, young fruiting body, fruiting body, and spores) of *P. ostreatus*. Figure 1B shows the changes in the NO content in different developmental stages of *P. ostreatus*. The results showed that the NO content in the mycelia of the culture bottles was significantly higher than that in the plates; this content was increased by 16.8-fold. After the formation of primordia, the NO content in the primordia decreased by 56.4% compared to that in the culture bottles. It is suggested that NO may play a negative role in primordia formation. After that, the content of NO in the young fruiting body and the fruiting body was 0.107 and 0.021 µmol/µg protein, respectively. The content of NO in spores increased slightly to 0.237 µmol/µg protein.

### 3.2. NO Plays a Negative Role in Primordia Formation

In this study, an NO donor (SNP) and the NO scavenger, cPTIO, were added to verify the function of NO. Figure 2 shows that SNP affects the rapid formation of primordia, prolongs the time of primordia formation, and then affects the development cycle. However, it does not affect the formation of the fruiting body. There was no significant effect on the development of the fruiting body when cPTIO was added.

### 3.3. NO Inhibits ACO Enzyme Activity and Mitochondrial aco Gene Expression

There are two subtypes of ACO in organisms. The mitochondrial ACO (mACO) subtype is a component of the TCA cycle, and the cytoplasmic ACO (cACO) subtype participates in the glyoxylic acid cycle. Figure 3A,B show that the activity of cytoplasmic ACO and mitochondrial ACO increased significantly when the mushroom bottles were transferred to the mushroom production room for 48 h. This result suggests that the TCA cycle is hastened during primordia formation in *P. ostreatus*. Interestingly, the addition of an exogenous NO donor (SNP) can significantly inhibit the activity of the ACO enzyme in the mitochondria and cytoplasm of *P. ostreatus*. However, cPTIO completely blocked the effect of the NO donor on the activity of ACO. As shown in Figure 3C, mitochondrial ACO enzyme activity was 5.7-fold higher than the cytoplasmic ACO enzyme activity. Therefore, mitochondrial ACO was selected for further study. Figure 3D shows that compared to that of the full-bottle mycelia (CK), the expression level of the mitochondrial *aco* gene in the mycelia of CK (48 h) was significantly increased by 2.9-fold. The addition of exogenous SNP inhibited the expression of the *aco* gene, as it was decreased by 25.3%. In contrast, the addition of the NO scavenger cPTIO promoted the expression of the *aco* gene, as it was increased by 13.3%.

### 3.4. The aco Gene Is Involved in Primordia Formation

Figure 4A,B show the expression levels of *aco* genes and ACO proteins at different developmental stages. Figure 4A shows that the *aco* gene in *P. ostreatus* was significantly upregulated during its growth and development. Compared to the *aco* gene expression level in the mycelia stage, those in the stages of primordia, fruiting bodies, and spores were increased by 5.0-fold, 16.0-fold, and 33.0-fold, respectively. Since the assembly from gene expression to mature protein is a very complex process, the formation of the ACO protein in different developmental stages was detected by Western blot. The results are shown in Figure 4B. The results showed that the ACO protein was detected in primordia, young fruiting bodies, and fruiting bodies, but not in other developmental stages. It is speculated that ACO protein was not detected because of its low expression levels in mycelia. In spores, a high expression of the *aco* gene was detected at the mRNA level, but no visible amount of ACO protein was detected.

To further explore the function of the *aco* gene in the development of *P.ostreatus*, the WT, OE-*aco*, and RNAi-*aco* strains were used for cultivation experiments. As shown in Figure 4, there was no significant difference in the growth rate of the different strains on PDA plates. However, the results of the cultivation experiment showed that the different strains showed differences in the formation stages of mycelia and primordia. As shown in Figure 4C, after 26 days of inoculation, the primordium was formed in both the OE-*aco* and WT strains. However, in the RNAi-*aco* strains, no primordia formation was observed. When the cultivation experiment was carried out for 29 days, the RNAi-*aco* strains also formed the primordium. This finding shows that *aco* gene interference prolonged the time of primordium formation. Then, when the cultivation experiment lasted for 32 days, the WT strain and OE-*aco* strains formed mature fruiting bodies, and the RNAi-*aco* strains formed young mushrooms. It can be concluded that RNAi of the *aco* gene can prolong the developmental cycle of the fruiting body.

### 3.5. aco Gene Interference Affects Energy Metabolism and Regulates H_2_O_2_ Production and Accumulation

Mitochondrial ACO is an important TCA cycle enzyme. In this study, mitochondrial *aco*-transformed strains were used to further explore the function of this gene in *P. ostreatus*. The culture bottles were transferred to the mushroom production room for 48 h, and then, mycelial samples of different strains were collected to detect the effects of the *aco* gene on the TCA cycle and energy metabolism (Figure 5). ICDHm and α-KGDH are key enzymes in the TCA cycle. Figure 5A,B show that *aco* OE can significantly increase the activities of ICDHm and α-KGDH, whereas RNAi-*aco* strains showed the opposite effect. This suggests that OE of *aco* can accelerate the TCA cycle.

Nicotinamide adenine dinucleotide exists in oxidized (NAD^+^) and reduced (NADH) forms in cells, and can participate in cell energy metabolism [35]. As shown in Figure 5C,D, the average NADH content in the OE-*aco* strains was 2.6-fold higher than that of the WT strain, and the average NAD^+^/NADH ratio in these strains decreased by 67.8%. In contrast, the average NADH content of the RNAi-*aco* strains decreased by 44.5% compared to that in the WT strain, and the NAD^+^/NADH ratio increased by 47.0%. Therefore, it is further speculated that OE of the *aco* gene can increase NADH content in the TCA cycle.

These small molecules of NAD participate in various biological processes, including energy metabolism, redox metabolism, and biosynthesis [36]. Figure 5E shows that during the formation of primordia, the ATP content in OE-*aco* strains increased slightly, with an average increase of 8.4%. The ATP content in *aco*-interfering strains decreased by 30.5%. Interestingly, the significant upregulation of NADH content in OE-*aco* strains did not cause significant changes in ATP content. Figure 5F shows that the H_2_O_2_ content in OE-*aco* strains was significantly upregulated, and the H_2_O_2_ content in the OE-*aco* 18 and OE-*aco* 2 strains was upregulated by 36.2% and 59.4%, respectively. These findings are consistent with the change trend of the NADH content in OE-*aco* strains. The H_2_O_2_ content in RNAi-*aco* 76 and RNAi-*aco* 15 decreased by 24.5% on average.

Therefore, OE of *aco* can further regulate the TCA cycle, affect energy metabolism, and promote H_2_O_2_ accumulation.

### 3.6. H_2_O_2_ Plays an Important Role in P. ostreatus Development

H_2_O_2_ has many biological functions as a signaling molecule [37]. In this study, H_2_O_2_ was detected in different developmental stages of *P. ostreatus*, and the content of H_2_O_2_ showed a specific change trend (Figure 6A,B). Previous studies have shown that H_2_O_2_ is abundant at the edge of colonies on PDA plates. It is speculated that the presence of H_2_O_2_ may regulate the elongation of apical mycelia [38]. The results of this study showed that H_2_O_2_ accumulated gradually during the transformation from mycelia to primordia in *P. ostreatus* (Figure 6A,B). Figure 6B shows that in the mycelial, primordial, and young fruiting body stages, the content of H_2_O_2_ increased significantly. It started to decrease in the fruiting body stage, and the content was the lowest in the spores.

### 3.7. H_2_O_2_ Promotes the Rapid Formation of Primordia

In this study, different concentrations of H_2_O_2_ and H_2_O_2_ scavenger (DMTU) were added to explore the function of H_2_O_2_ in *P. ostreatus*. Figure 7 shows that 26 days after inoculation, the experimental group with exogenous addition of 25 mM and 50 mM H_2_O_2_ formed primordia faster than the control group. Therefore, it is speculated that the addition of exogenous H_2_O_2_ promoted the transformation from vegetative growth to reproductive growth in *P. ostreatus*. In contrast, the rate of primordia formation was slowed down after the exogenous addition of DMTU. After 32 days of inoculation, mature fruiting bodies formed in the control group and the exogenous H_2_O_2_ addition experimental group. Unexpectedly, some *P. ostreatus* in the DMTU experimental group formed young fruiting bodies, and some did not form primordia. It is speculated that the addition of the H_2_O_2_ scavenger seriously affected the formation of primordia.

## 4. Discussion

As an important and universal signaling molecule, NO can participate in a variety of developmental processes [39,40]. In plants, NO causes a delay in floral transition, which indicates that NO plays a negative role in the transformation from vegetative growth to reproductive growth [41]. Compared to plants, the function of NO in edible mushrooms has not been fully studied. In this study, the results showed that the content of NO decreased gradually during the transformation from vegetative growth to reproductive growth. It is speculated that NO may play a negative role in this process. Subsequently, experiments with exogenous NO donors and scavengers further confirmed that NO played a negative role in the primordium formation stage. The effects of NO found in this study are similar to those found in plants. In addition, studies in plants showed that NO regulates flowering through different pathways. Previous studies have shown that NO in plants regulates the activities of sugar metabolism enzymes through S-nitrosation, affects the synthesis of reducing polysaccharides, and reduces ATP levels [42,43]. It is speculated that NO is closely related to the regulation of energy metabolism. Additionally, ACO can participate in the regulation of cellular metabolism [44]. Previous studies have shown that mitochondrial ACO is one of the main targets of NO [45,46]. In this study, the addition of an exogenous NO donor (SNP) significantly inhibited the activities of cytoplasmic ACO and mitochondrial ACO enzymes in *P. ostreatus* during the transformation from vegetative growth to reproductive growth. Additionally, the expression of the mitochondrial ACO coding gene was also significantly inhibited (Figure 3), which shows that NO inhibits the gene expression and enzyme activity of mitochondrial ACO during the formation of the *P. ostreatus* primordium. This result is similar to previous research results, which further indicated that ACO is one of the targets of NO in *P. ostreatus* [19]. It is interesting that NO has a much greater effect on ACO activity than on the level of *aco* gene expression. Previous studies have shown that in *Ganoderma lucidum*, NO can directly act on ACO through S-nitrosation [46]. It is speculated that in *P. ostreatus*, NO can directly affect ACO enzyme activity. However, as a signaling molecule, NO regulates the expression of nuclear genes, which is a complex regulatory network.

The expression patterns of ACO were different in different developmental stages. The results showed that in *Arabidopsis thaliana*, the expression of the *aco* gene was low in most developmental stages, but significantly increased in seeds and pollen [47]. This study showed that the relative expression of ACO in *P. ostreatus* at different developmental stages was also different. Interestingly, although the mitochondrial *aco* gene was abundantly expressed in spores, a large number of proteins were not detected at this stage. According to the literature, mitochondrial ACO has two major functions in eukaryotes: as an enzyme in the TCA cycle, and as a biosensor for ROS and iron [48,49]. Therefore, it is speculated that the mRNA of *aco* transcription may not be translated into mature ACO proteins in spores, and its specific biological function needs further study.

Recently, genetic transformation technology has been used in edible fungi [5,50]. To explore the regulatory pathway by which NO inhibits ACO to slow down the rate of primordium formation, mushroom experiments were carried out with OE-*aco* and RNAi-*aco* strains. At present, research on *aco* function in microorganisms is also gradually increasing. For example, in *Staphylococcus aureus*, the inactivation of ACO inhibited growth after the exponential period, and increased the survival rate during the stable period [51]. In *Streptomyces coelicolor*, the *aco* disruption mutant did not grow on minimal glucose media in the absence of glutamate [52]. In *Xanthomonas campestris*, the loss of *aco* affects the transcription and activity of some extracellular enzymes, and reduces the amount of extracellular polysaccharide produced by the organism [53]. A *Streptomyces viridochromogenes* Tü494 strain expressing a mutant *aco* gene cannot form any aerial mycelia, spores, or phosphinothricin tripeptide [27]. However, in *P. ostreatus*, RNAi of the mitochondrial *aco* gene did not affect the formation of aerial mycelia. However, it did affect the time of primordia formation and prolong the developmental cycle of fruiting bodies. The reason for this phenomenon might be that RNAi only reduce gene expression level, rather than disrupt the specific genes. In addition, the results of this study showed that the *aco* gene and ACO protein expression levels were significantly increased in the primordia compared with the mycelia stage (Figure 4A,B). It is speculated that ACO plays an important role in primordia formation. Therefore, the OE-*aco* and RNAi-*aco* strains showed differences in the time of primordia formation. It can be inferred that there are differences in the function of mitochondrial *aco* in different organisms. Furthermore, ACO is involved in a variety of abiotic stress response pathways in plants, such as heat stress [54] and oxidative stress [55]. In the process of *P. ostreatus* cultivation, the mycelium needs to be stimulated by low temperatures to form the primordium [56]. In other words, low temperature stimulation is equivalent to cold stress for the mycelium. Moreover, an increase in *aco* gene expression in the primordia was detected. It is suggested that low temperature stimulation may affect the upregulation of *aco* gene expression in the primordium, and then promote the differentiation of the primordium. Moreover, the *aco* gene extended the time of primordium formation after interference, which further suggested that the *aco* gene participated in the development of *P. ostreatus*.

The TCA cycle is distributed throughout mitochondria, and is the final metabolic pathway and connection hub of sugars, lipids, and amino acids [57]. It is closely related to energy metabolism [58]. Considering that ACO is the key enzyme in the TCA cycle, it is speculated that ACO gene interference may reduce the TCA cycle rate and affect the metabolism of carbohydrates, lipids, and amino acids. Thus, it can reduce energy metabolism and affect the development of the fruiting body. ICDHm and α-KGDH are the rate-limiting enzymes in the TCA cycle [59]. In this study, the activity of ICDHm and α-KGDH was significantly increased in the OE-*aco* strains, and the opposite results were observed in the RNAi-*aco* strains. Additionally, the interference of *aco* also significantly reduced the content of NADH and ATP. In plants, studies have shown that NO can inhibit ATP synthase activity and reduce ATP synthesis through S-nitrosation [42]. The current results were similar to those in plants. In a word, NO can regulate ATP production in plants and edible fungi, but its regulatory pathway is different.

H_2_O_2_ has a variety of signaling functions, which are very important for the growth and development of plant systems [60,61]. In this study, H_2_O_2_ accumulated gradually during the transformation from mycelia to primordia in *P. ostreatus*, which may be due to exposure to cold stress and light stress in the mushroom room. It is suggested that H_2_O_2_ plays an important role in the transformation from vegetative growth to reproductive growth in *P. ostreatus*. Further results showed that *aco* OE or RNAi could also regulate the production and content of H_2_O_2_. In plants, H_2_O_2_ can not only inhibit the development of leaves, but can also enhance the expression of flower-related genes. Thus, it can promote plant reproductive development [62]. The current results were similar to those in plants, indicating that H_2_O_2_ can play an active regulatory role in reproductive growth as a signaling molecule.

## 5. Conclusions

The current data suggested that during the growth and development of *P. ostreatus*, NO signaling molecules can negatively regulate the rapid formation of primordia by inhibiting ACO, affecting the TCA cycle rate, reducing the production of NADH, and affecting the contents of ATP and H_2_O_2_ (Figure 8). This study preliminarily analyzed the function and regulatory pathway of NO in the growth and development of *P. ostreatus*. In the future, the mechanism by which H_2_O_2_ regulates the formation of the *P. ostreatus* primordium will be further studied.

## Figures and Tables

**Figure 1 jof-08-01055-f001:**
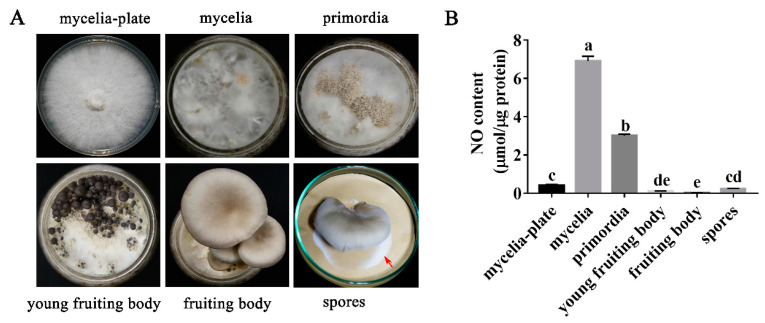
Determination of the NO content at different developmental stages of *P. ostreatus*. (**A**) Different developmental stages of *P. ostreatus* (spores: the position pointed by the red arrow). (**B**) NO content in *P. ostreatus* at different developmental stages. Different letters in (**B**) indicate significant differences for the comparison of samples (*p* < 0.05 according to Duncan’s test).

**Figure 2 jof-08-01055-f002:**
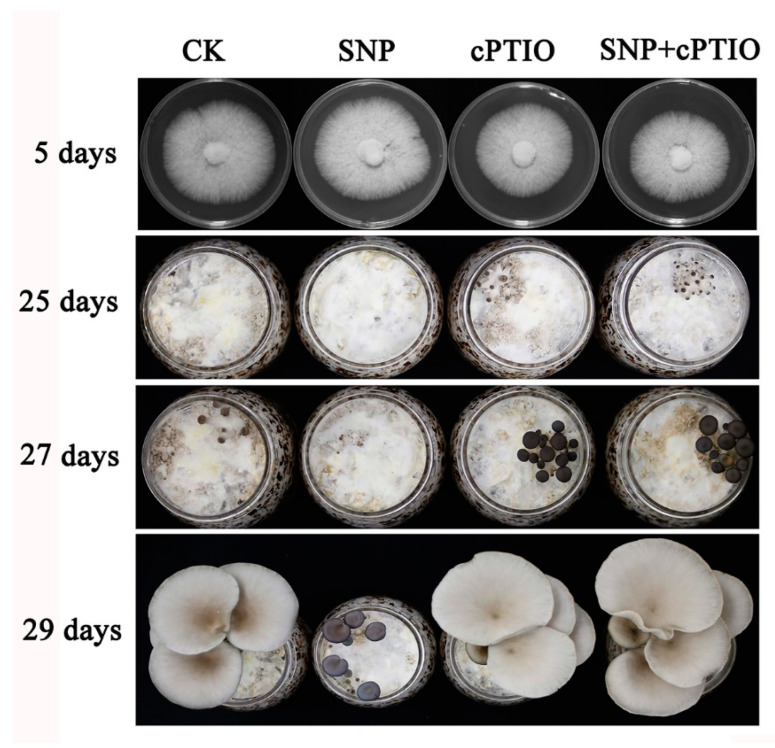
The addition of exogenous NO influences the formation of primordia.

**Figure 3 jof-08-01055-f003:**
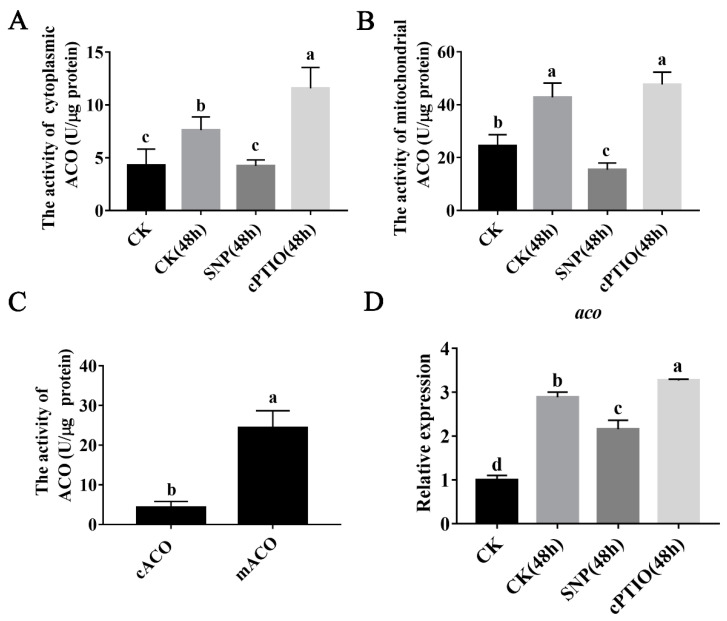
Exogenous NO inhibited *aco* gene expression and ACO activity in primordia formation. (**A**) ACO activity detected in the cytoplasm of mycelia. (**B**) ACO activity detected in the mitochondria. (**C**) Comparison of ACO activity in the cytoplasm and mitochondria of mycelia at 48 h after transferring mushroom bottles to a mushroom production room. (**D**) Expression of the mitochondrial *aco* gene. Different letters indicate significant differences for the comparison of samples (*p* < 0.05 according to Duncan’s test).

**Figure 4 jof-08-01055-f004:**
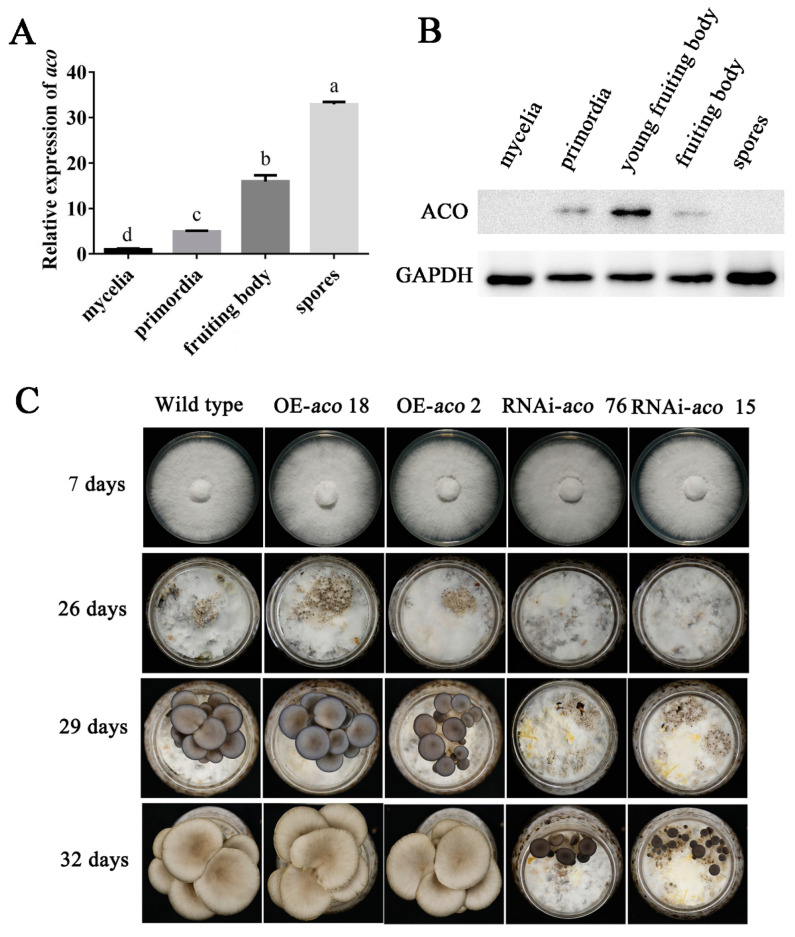
The *aco* gene is involved in primordia formation. (**A**) Expression level of the *aco* gene at different growth stages. (**B**) Protein expression of *aco* at different growth stages. (**C**) Developmental stages in the WT, OE, and RNAi-*aco* strains. Different letters in (**A**) indicate significant differences for the comparison of samples (*p* < 0.05 according to Duncan’s test).

**Figure 5 jof-08-01055-f005:**
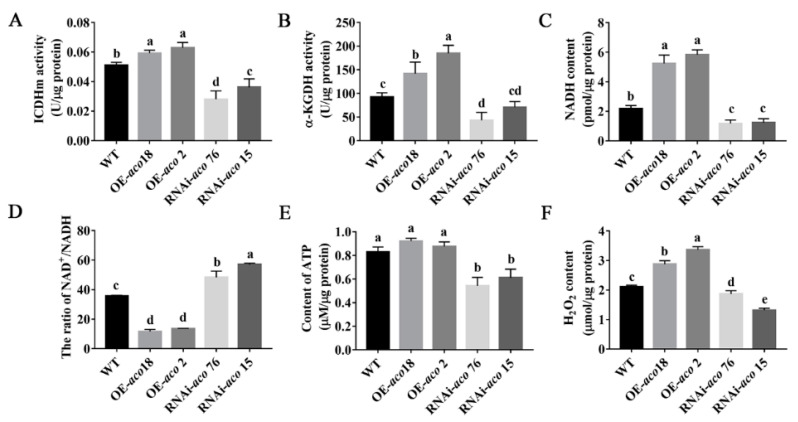
The *aco* gene affects the TCA cycle and energy metabolism. (**A**) Detection of ICDHm activity in *aco*-transformed strains. (**B**) Detection of α-KGDH activity in *aco*-transformed strains. (**C**) NADH content. (**D**) NAD^+^/NADH ratio. (**E**) ATP content. (**F**) H_2_O_2_ content. Different letters indicate significant differences for the comparison of samples (*p* < 0.05 according to Duncan’s test).

**Figure 6 jof-08-01055-f006:**
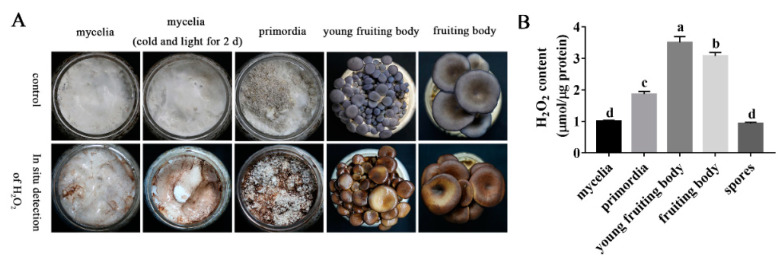
H_2_O_2_ plays an important role in the development of *P. ostreatus*. (**A**) Detection of H_2_O_2_ in different developmental stages. (**B**) Detection of H_2_O_2_ content in different developmental stages of *P. ostreatus*. Different letters in (**B**) indicate significant differences for the comparison of samples (*p* < 0.05 according to Duncan’s test).

**Figure 7 jof-08-01055-f007:**
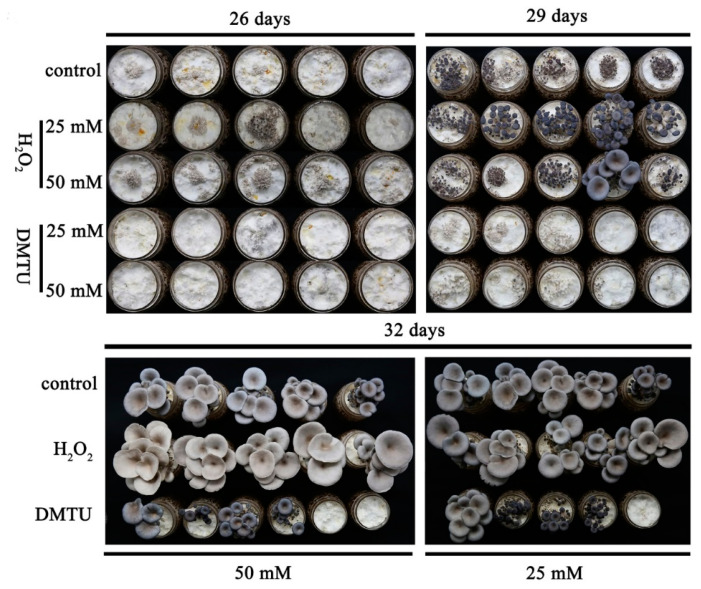
The addition of exogenous H_2_O_2_ affected the formation time of primordia.

**Figure 8 jof-08-01055-f008:**
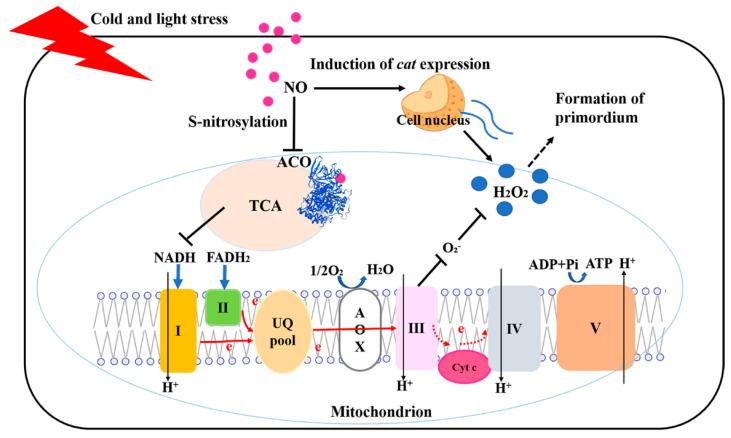
Schematic representation of NO regulation of the growth and development of primordia via the regulation of H_2_O_2_.

**Table 1 jof-08-01055-t001:** Primers used in this study.

Primer	Sequence (5′→3′)	Note
qPCR_*β-actin*-F	AGTCGGTGCCTTGGTTAT	qPCR
qPCR_*β-actin*-R	ATACCGACCATCACACCT	
qPCR_*aco*-F	CCTCACCGTTCTCAATGTT	
qPCR_*aco*-R	CGACGAAGGCATGAGTAG	

## Data Availability

All data generated or analyzed during this study are included in this published article.

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
