# Peer review of "Nitric Oxide Negatively Regulates the Rapid Formation of *Pleurotus ostreatus* Primordia by Inhibiting the Mitochondrial *aco* Gene"

_jof, 2022, doi:10.3390/jof8101055_

Round 1

Reviewer 1 Report

The current study reports an interesting topic that points out the negative regulation of nitric oxide on the rapid formation of Pleurotus ostreatus primordia by inhibiting the mitochondrial aco gene. The manuscript shows originality and novelty but needs major adjustments in its standard English. Therefore, I invite the authors to pass their manuscript to a native English speaker for editing and revision. The presented parts are significant and interpreted appropriately. The raised conclusions and further suggestions are justified. The study covers its topic which is well relevant and all used references (except some in the Discussion part) are appropriate. Also, the study is correctly designed and sounds technically.

The Abstract part is aiming and clear. The mention of overall methodology adopted is a must in this part. The avoidance of the first voice form of the sentence is sympathetically recommended scientifically and the use of the impersonal form is very appropriate. On the other hand, all keywords fit. The Introduction part is well structured and aiming. It needs very minor adjustments in terms of linguistic mistakes. However, the study’s problematic and aims are very clear and interesting. The Materials and methods part is well structured and clear. However, several methods should be explained briefly; whereas, others are appropriately mentioned and are well clear for further repetition by other researchers. Some sentences should be reformulated in a more appropriate standard English besides other minor linguistic adjustments. Some statements lack reliable sources (references) that should be provided accordingly. The Results part is aiming. The scientific analysis of the findings was performed in a good manner but can be improved. A correct statistical approach was performed adequately. However, this part should be merged with the Discussion part or numerous statements (of which some are presented below) should be moved to the latter as they are inappropriate in the Results part. Moreover, the use of the impersonal form of the sentence rather than the first voice’s one is a must. Additionally, numerous statements lack reliable sources (references) that should be provided accordingly. Besides, several linguistic adjustments are sympathetically suggested. The Discussion part needs to be merged to the Results part as large sections show duplication with the previous part. Numerous sentences should be reformulated in the impersonal form rather than the first voice’s one, in a less cumbersomeness and more appropriate linguistically. Some sections should be removed as going a little bit out of the study’s subject. Several statements lack reliable sources (references) that should be provided. Other minor linguistic adjustments are also suggested. However, the majority of mentioned sources are reliable and directly related to the study’s findings. The Conclusions part is a well-structured and aiming one. It summarizes appropriately the findings of the study and suggests further related research. Minor adjustments related to sentence reformulation in the impersonal form rather than the first voice’s one and other linguistic mistakes are only needed in this part.

Briefly, based on the above and below detailed explanation, the manuscript needs adjustments but shows a merit to be published in “Journal of Fungi” once all suggestions and recommendations are fully addressed.

Abstract

1)      Page 1, lines 13–29: The Abstract part is aiming and clear. The mention of overall methodology adopted is a must in this part. The avoidance of the first voice form of the sentence is sympathetically recommended scientifically and the use of the impersonal form is very appropriate. On the other hand, all keywords fit.

2)      Page 1, line 15: Kindly mention the adopted methodology in the study before outlining the findings.

3)      Page 1, lines 15–17: “In this… Pleurotus ostreatus”: Kindly avoid the first voice form of the sentence and adopt the impersonal form instead.

4)      Page 1, lines 30–31: All keywords fit well.

1. Introduction

1)      Pages 1–2, lines 34–98: The Introduction part is well structured and aiming. It needs very minor adjustments in terms of linguistic mistakes. However, the study’s problematic and aims are very clear and interesting.

2)      Page 1, line 37: Kindly remove “Recently” to avoid the extensive mention of this term.

3)      Page 2, line 63: Kindly remove “also”.

4)      Page 2, line 66: Kindly remove “our”.

5)      Page 2, lines 75–76: “In iron… clusters”: The sentence is badly written in standard English; accordingly, kindly reformulate it.

6)      Page 2, line 78: Kindly replace “but” by “whereas”.

7)      Page 2, lines 83–84: Kindly adjust as follow: “affects”.

8)      Page 2, line 85: Kindly adjust as follow: “decreases”.

2. Materials and methods

1)      Pages 3–4, lines 100–170: The Materials and methods part is well structured and clear. However, several methods should be explained briefly; whereas, others are appropriately mentioned and are well clear for further repetition by other researchers. Some sentences should be reformulated in a more appropriate standard English besides other minor linguistic adjustments. Some statements lack reliable sources (references) that should be provided accordingly.

2)      2.2. Mushroom Production: Page 3, line 111: Kindly mention herein that you left the bottles to cool down at room temperature in sterilized conditions.

3)      2.3. Determination of the NO content: Page 3, line 117: Kindly adjust as follow: “were placed”.

4)      2.3. Determination of the NO content: Page 3, line 118: Kindly adjust as follow: “were added”.

5)      2.4. Experiment with the addition of exogenous SNP or cPTIO: Page 3, lines 122–124: “In the mushroom… group”: The sentence is badly written in standard English; accordingly, kindly reformulate it.

6)      2.4. Experiment with the addition of exogenous SNP or cPTIO: Page 3, line 125: Kindly adjust as follow: “added”.

7)      2.6. ICDHm and ɑ-KDGH Activity Determination: Page 3, lines 134–139: “ICDHm… cycle”: These statements lack reliable sources (references); accordingly, kindly provide them.

8)      2.9. Western Blot Analysis: Page 4, line 149: Kindly replace “our” by “a”.

9)      2.9. Western Blot Analysis: Page 4, line 152: Kindly explain briefly the adopted methodology.

10)  2.10. Quantitative Real-Time PCR (qPCR): Page 4, lines 154–155: Kindly explain briefly the adopted methods.

11)  2.11. Experiment with the Addition of Exogenous H2O2 or N,N'-dimethylthiourea (DMTU): Page 4, lines 160–162: “In the mushroom… group”: The sentence is badly written in standard English; accordingly, kindly reformulate them.

12)  2.12. Data analysis: Page 4, line 168: Kindly mention the version of SPSS program, the company and its location.

13)  2.12. Data analysis: Page 4, line 168: Kindly replace “are” by “were”.

3. Results

1)      Pages 4–11, lines 172–329: The Results part is aiming. The scientific analysis of the findings was performed in a good manner but can be improved. A correct statistical approach was performed adequately. However, this part should be merged with the Discussion part or numerous statements (of which some are presented below) should be moved to the latter as they are inappropriate in the Results part. Moreover, the use of the impersonal form of the sentence rather than the first voice’s one is a must. Additionally, numerous statements lack reliable sources (references) that should be provided accordingly. Besides, several linguistic adjustments are sympathetically suggested.

2)      3.1. NO Content in P. ostreatus Varies in Different Developmental Stages: Page 4, lines 174–175: “In this… stages”: Kindly avoid the first voice form of the sentence and adopt the impersonal form instead.

3)      3.1. NO Content in P. ostreatus Varies in Different Developmental Stages: Page 4, line 180: Kindly adjust as follow: “compared to”.

4)      3.1. NO Content in P. ostreatus Varies in Different Developmental Stages: Page 4, line 184: Kindly remove “In conclusion” and replace it by “Therefore”.

5)       3.2. NO Plays a Negative Role in Primordia Formation: Page 5, line 195: Kindly remove “(Figure 2)”.

6)      3.3. NO Inhibits ACO Enzyme Activity and Mitochondrial aco Gene Expression: Page 6, line 207: Kindly replace “accelerating” by “hastened”.

7)      3.3. NO Inhibits ACO Enzyme Activity and Mitochondrial aco Gene Expression: Page 6, lines 212–213: Kindly adjust as follow: “compared to”.

8)      3.4. The aco Gene is Involved in Primordia Formation: Page 7, line 229: Same recommendation as in the previous comment.

9)      3.4. The aco Gene is Involved in Primordia Formation: Page 7, line 240: Kindly remove “In conclusion” and replace it by “Therefore”.

10)  3.4. The aco Gene is Involved in Primordia Formation: Page 7, lines 246–247: “However… primordia”: Kindly avoid the first voice form of the sentence and adopt the impersonal form instead.

11)  3.4. The aco Gene is Involved in Primordia Formation: Page 7, line 258: Kindly remove “In conclusion” and replace it by “Therefore”.

12)  3.5. aco Gene Interference Affects Energy Metabolism and Regulates H2O2 Production and Accumulation: Page 8, lines 267–269: “The TCA… enzyme”: First, these statements lack reliable sources (references); accordingly, kindly provide them. Second, kindly move these sentences to the Discussion part.

13)  3.5. aco Gene Interference Affects Energy Metabolism and Regulates H2O2 Production and Accumulation: Page 8, lines 272–273: Kindly replace “and” by “while” and adjust as follow: “showed”.

14)  3.5. aco Gene Interference Affects Energy Metabolism and Regulates H2O2 Production and Accumulation: Page 8, lines 275–276: “Nicotinamide… metabolism”: This statement lacks reliable sources (references); accordingly, kindly provide them.

15)  3.5. aco Gene Interference Affects Energy Metabolism and Regulates H2O2 Production and Accumulation: Page 9, line 279: Kindly remove “was”.

16)  3.5. aco Gene Interference Affects Energy Metabolism and Regulates H2O2 Production and Accumulation: Page 9, line 280: Kindly adjust as follow: “compared to” and remove “was”.

17)  3.5. aco Gene Interference Affects Energy Metabolism and Regulates H2O2 Production and Accumulation: Page 9, line 283: “NADH… cells”: This statement lacks reliable sources (references); accordingly, kindly provide them.

18)  3.5. aco Gene Interference Affects Energy Metabolism and Regulates H2O2 Production and Accumulation: Page 9, lines 287–288: “The formation… ROS”: Same recommendation as in the previous comment. Moreover, kindly move this statement to the Discussion part.

19)  3.5. aco Gene Interference Affects Energy Metabolism and Regulates H2O2 Production and Accumulation: Page 9, line 298: Kindly remove “In conclusion” and replace it by “Therefore”.

20)  3.6. H2O2 Plays an Important Role in P. ostreatus Development: Page 9, line 301: “H2O2… molecule”: This statement lacks reliable sources (references); accordingly, kindly provide them.

21)  3.6. H2O2 Plays an Important Role in P. ostreatus Development: Page 9, lines 305–308: “Moreover… P. ostreatus”: Same recommendation as in the previous comment. Moreover, kindly move these statements to the Discussion part.

22)  3.6. H2O2 Plays an Important Role in P. ostreatus Development: Page 9, lines 309–311: “The results… spores”: Kindly reformulate this sentence using the past tense.

23)  3.7. H2O2 Promotes the Rapid Formation of Primordia: Page 10, line 322: Kindly adjust as follow: “was slowed down”.

4. Discussion

1)      Pages 11–13, lines 331–419: The Discussion part needs to be merged to the Results part as large sections show duplication with the previous part. Numerous sentences should be reformulated in the impersonal form rather than the first voice’s one, in a less cumbersomeness and more appropriate linguistically. Some sections should be removed as going a little bit out of the study’s subject. Several statements lack reliable sources (references) that should be provided. Other minor linguistic adjustments are also suggested. However, the majority of mentioned sources are reliable and directly related to the study’s findings.

2)      Page 11, line 332: Kindly adjust as follow: “Compared to”.

3)      Page 11, lines 333–335: “In this… growth”: Kindly avoid the first voice form of the sentence and adopt the impersonal form instead.

4)      Page 11, line 337: Kindly remove “also”.

5)      Page 11, line 338: Kindly adjust as follow: “showed that”.

6)      Page 11, lines 344–347: “In this… growth”: Kindly avoid the first voice form of the sentence and adopt the impersonal form instead.

7)      Page 11, line 350: Kindly remove “our”.

8)      Page 11, line 351: Kindly adjust as follow: “indicated”.

9)      Page 11, line 352: “This is… results”: Kindly remove this sentence and move the in-text reference “[27]” to the end of the previous sentence.

10)  Page 12, lines 354–355: “The results… pollen”: This statement lacks reliable sources (references); accordingly, kindly provide them.

11)  Page 12, lines 355–356: “Our… predecessors”: Kindly remove this sentence.

12)  Page 12, line 356: Kindly replace “is” by “was”.

13)  Page 12, lines 365–367: “The structure… depth”: These statements lack reliable sources (references); accordingly, kindly provide them.

14)  Page 12, lines 367–369: “For example… viability”: The sentence is cumbersome; accordingly, kindly reformulate in order to make it clearer and more aiming.

15)  Page 12, lines 365–375: “The structure… activities”: This section is out of subject; accordingly, kindly remove it as you cannot compare plants to animals (there is a wide difference).

16)  Page 12, lines 375–381: “In this… formation”: This is a repetition of the results; accordingly, kindly refer to the first comment in the Results part comments.

17)  Page 12, line 381: Kindly adjust the sentence as follow: “The current results showed that…” and remove “also”.

18)  Page 12, lines 391–393: “In the process… primordium”: This statement lacks reliable sources (references); accordingly, kindly provide them. You can use the following study as a reliable reference: https://doi.org/10.176660/ActaHortic.2021.1327.32.

19)  Page 12, lines 394–395: “We also… primordium”: Kindly avoid the first voice form of the sentence and adopt the impersonal form instead.

20)  Page 12, lines 401–402: “ICDHm… cycle”: This statement lacks reliable sources (references); accordingly, kindly provide them.

21)  Page 12, line 406: Kindly adjust the sentence as follow: “The current results were similar…”

22)  Page 13, lines 416–417: “Furthermore… [54]”: Kindly remove this sentence.

23)  Page 13, line 417: Kindly adjust the sentence as follow: “The current results were similar…”

5. Conclusions

1)      Page 13, lines 421–428: The Conclusions part is a well-structured and aiming one. It summarizes appropriately the findings of the study and suggests further related research. Minor adjustments related to sentence reformulation in the impersonal form rather than the first voice’s one and other linguistic mistakes are only needed in this part.

2)      Page 13, line 421: Kindly remove “In conclusion”.

3)      Page 13, line 422: Kindly remove the in-text reference from the end of the sentence as mentioning references in the Conclusions part is often not very appropriate scientifically.

4)      Page 13, line 422: Kindly remove “In this study” and adjust the sentence as follow: “The current data suggested that…”

5)      Page 13, line 426: Kindly adjust as follow: “This study”.

6)      Page 13, lines 427–428: “In the future… primordia”: Kindly avoid the first voice form of the sentence and adopt the impersonal form instead.

Author Response

请参阅附件

Reviewer 2 Report

The authors have analysed the effect of NO on the formation of primordia in P. ostreatus. They either added an NO donor or an NO scavenger to measure its effect of the primordia formation.
The title of the manuscript suggest that NO has a direct effect on primordia formation in P. ostreatus. The manuscript shows, however, clearly that NO inhibits the expression of the Aco gene and thus interferes with the TCA cycle. As a results, the production of NADH and ATP is negatively affected. It is no wonder that this reduction in energy production inhibits a number of physiologic processes including the development of mushrooms. It is also not clear in the manuscript if NO only affects the activity of Aco or might affect also other enzymes in the TCA cycle.

There are also number of items in the methods used that needs clarification in the manuscript.

·        It is not clear how exogenous SNP, cPTIO and H2O2 were applied in the bottle production. I assume that is was applied on top of the substrate. It is difficult to imagen how to distribute 1 ml on a hydrophobic surface. And how do you remove the excess after 24 hours of incubation? I assume that after 25 hours nothing is left of the 1 ml added.

·        Lysate was used in the determination of intracellular NO. What lysate was used and does it work at 0 oC?

·        In line 109 there is a reference (29) to how culture medium was prepared. There is, however, no culture description in this reference or substrate description for the bottle cultivation. In addition, the DOI link of this reference is not correct.

·        In line 134 the authors state that ICDHm catalyzes the formation of isocitrate. This enzyme catalyzes the formation of α-keto-glutarate.

·        The NO concentration and the enzyme activities in the mycelium on plate or on the substrate are expressed per unit of protein. How was the mycelium from an agar plate and the substrate isolated for this measurement? I assume that in an agar plate (potato dextrose agar) and the substrate also contain protein that might interfere with the correct measurement of NO per unit protein. Please clarify.

·        Aco activities were measured in the cytoplasm and in mitochondria. How was the discrimination done between the activity of a similar enzyme in different cell compartments?

Other items:

Line 34: “..heterologous edible fungi..”. What do the authors mean with “heterologous”?
Line 86-87: Move these sentences up before line 70 which makes clear why the authors want to study Aco expression and activity.
Line 107-108: What does "activated"  mean here? Is it transferred to new media?
Line 157: “..were..” change to “..are..”
Line 179 and 180: “..16.679 fold”  and “..56.358%..”. This suggest an accuracy that is not realistic. Use no or one decimal.
Line 184-184: Move to discussion.
Line 193: “..could affect..”; “..affects..”.
Line 194-195: The addition of SNP prolongs the time of primordia formation. Does that mean that if incubated longer in the end primordia and mushrooms will be formed? And how much later that in the control?
Line 196-197: Move to discussion.
Line 204: Figure 3A: Indicate in Figure 3 that 48H indicate the time after the fully colonized bottles were transferred to the mushroom production unit and mention in M&M what the climate change is in this room compared to the previous period. The legend indicates Aco enzyme activities during primordia formation. 48 H after transfer to the production room no primordia are formed to my knowledge. In figure 3C: When at what time are these activities compared (0 h or 48 h)? Correct the legend (“..detcsted..”).
Line 214-215: The effect of SNP is much larger on the activity of Aco than on the expression level of the Aco gene. Discuss this in the Discussion paragraph.
Lines 218-219: Move to Discussion.
Line 232: “..the expression of the ACO protein..”; Should be “.. the formation of..”
Line 237: “..large amount..”; “..no visible amount..”
Lines 238-242: Move to Discussion.
Lines 238-242: Aco plays an important role in the TCA cycle and the generation of energy. Seeing no effect on mycelial growth on plates, what does this mean? Is the high nutrient content of type of nutrient of these plates the cause of a lack of the effect? Please discuss this in the Discussion paragraph.
Lines 248-249: Replace primordium with primordia.
Lines 255-260: Move to Discussion.
Figure 5: There is no information here at what stage the activities were measured.
Line 303: “..apical mycelium..”. Apical is used to refer to tips of hyphae. I assume that the authors refer here to the edge of the colony? Or has microscopic analyses shown that indeed apices were the location of high concentrations of H2O2?
Lines 305-308: Move to Discussion.
Lines 376-379: Overexpression of Aco has an effect on enzyme activities in the TCA cycle but there seems to be no phenotypic effect. Why is that? If Aco is essential in primordia formation I would expect an acceleration of primordia formation in overexpressed mutants.
Lines 385-386: I have no idea why the presence of 2 nuclei in P. ostreatus explains the absence of inhibition of Aco on the formation of areal hyphae.

Reviewer 3 Report

I reviewed your manuscript and I are very satisfied for your results, howover, You should correct some aspect in your document, I send the file in pdf with any comentars of this work.

Round 2

Reviewer 1 Report

Authors made significant improvements to their manuscript and are well thanked for that. Only minor suggestions are still raised and presented below.

Briefly, based on the overall manuscript’s evaluation, it needs some adjustments but shows now a high merit to be published in “Journal of Fungi” once all suggestions and recommendations are fully addressed.

2. Materials and methods

1)      2.4. Experiment with the Addition of Exogenous Sodium Nitroprusside (SNP) or 2-(4-carboxyphenyl)-4,4,5,5-tetramethylimidazoline-1-oxyl-3-oxide (cPTIO): Page 3, lines 126–127: Kindly adjust as follow: “10 bottles per group”.

2)      2.4. Experiment with the Addition of Exogenous Sodium Nitroprusside (SNP) or 2-(4-carboxyphenyl)-4,4,5,5-tetramethylimidazoline-1-oxyl-3-oxide (cPTIO): Page 3, lines 127–128: “When the culture… production”: The sentence is badly written in standard English; accordingly, kindly reformulate it.

3)      2.7. Measurement of the Activities of Isocitrate Dehydrogenase of Mitochondria (ICDHm) and α-ketoglutarate Dehydrogenase (α-KGDH): Page 4, line 150: Kindly adjust as follow: “[30,31]”.

4)      2.10. Western Blot Analysis: Page 4, lines 164–165: Kindly adjust as follow: “were separated”.

5)      2.12. Experiment with the Addition of Exogenous H2O2 or N,N'-dimethylthiourea (DMTU): Page 4, line 180: Kindly adjust as follow: “10 bottles per group”.

6)      2.12. Experiment with the Addition of Exogenous H2O2 or N,N'-dimethylthiourea (DMTU): Page 4, lines 180–182: “When the culture… carried out”: The sentence is badly written in standard English; accordingly, kindly reformulate it.

4. Discussion

1)      Page 12, lines 386–388: “It is… expression”: The sentence is badly written in standard English; accordingly, kindly reformulate it.

2)      Page 12, lines 395–397: “In the process… primordium”: You can also use the following study as a reliable reference: “Effect of olive pruning residues on substrate temperature and production of oyster mushroom (Pleurotus ostreatus, Acta Horticulturae, 1327, 245–252).
